# Non-Destructive Inspection of Physicochemical Indicators of Lettuce at Rosette Stage Based on Visible/Near-Infrared Spectroscopy

**DOI:** 10.3390/foods13121863

**Published:** 2024-06-13

**Authors:** Wei Li, Qiaohua Wang, Yingli Wang

**Affiliations:** 1College of Engineering, Huazhong Agricultural University, Wuhan 430070, China; liwei99ffff@163.com (W.L.); yingliwang@mail.hzau.edu.cn (Y.W.); 2Key Laboratory of Agricultural Equipment in the Middle and Lower Reaches of the Yangtze River, Ministry of Agriculture, Wuhan 430070, China

**Keywords:** lettuce, visible/near-infrared spectrum, physicochemical indicators, non-destructive inspection

## Abstract

Lettuce is a globally important cash crop, valued by consumers for its nutritional content and pleasant taste. However, there is limited research on the changes in the growth indicators of lettuce during its growth period in domestic settings. Quality assessment primarily relies on subjective evaluations, resulting in significant variability. This study focused on hydroponically grown lettuce during the rosette stage and investigated the patterns of changes in the indicators and spectral curves over time. By employing spectral preprocessing and selecting characteristic wavelengths, three models were developed to predict the indicators. The results showed that the optimal model structures were S_G-UVE-PLSR (SSC and vitamin C) and Nor-CARS-PLSR (moisture content). The PLSR models achieved prediction set correlation coefficients of 0.8648, 0.8578, and 0.8047, with residual prediction deviations of 1.9685, 1.9568, and 1.6689, respectively. The optimal models were integrated into a portable device, using real-time analysis software written in Matlab2021a, for the prediction of the physicochemical indicators of lettuce during the rosette stage. The results demonstrated prediction set correlation coefficients of 0.8215, 0.8472, and 0.7671, with root mean square errors of prediction of 0.5348, 1.5813, and 2.3347 for a sample size of 180. The small discrepancies between the predicted and actual values indicate that the developed device can meet the requirements for real-time detection.

## 1. Introduction

Lettuce, also known as leaf lettuce, is a globally important vegetable crop that originated in Europe and is now widely cultivated worldwide [1,2]. As one of the most popular vegetables in the world, lettuce is not only loved for its pleasant taste but also for its low calorie, low fat, and low sodium content. It is rich in minerals such as calcium, iron and potassium, fiber, folate, vitamin C, and carotenoids, as well as other nutrients and bioactive compounds. Lettuce has been associated with various health benefits, such as cholesterol and blood pressure reduction and the prevention of heart rhythm disorders [3,4,5]. Therefore, it is of great significance to rapidly and accurately assess the quality of lettuce. Currently, there is limited research on changes in the growth indicators of lettuce in China, and the grading of lettuce quality lacks clear guidelines. It is often dependent on subjective evaluations, leading to significant fluctuations in quality assessment, which has a significant impact on the development prospects of lettuce. Soluble solids content (SSC), moisture content, and vitamin C are three key physicochemical indicators of lettuce. SSC is composed of soluble sugars and organic acids, which play an important role in the flavor, texture, and marketability of lettuce. Moisture content is an important indicator of freshness, representing the ratio of water to dry matter in lettuce. The content of vitamin C can indirectly reflect the nutritional value of lettuce [6,7]. Currently, the commonly used methods for testing physicochemical indicators of fruits and vegetables involve destructive sampling. This method is labor-intensive, time-consuming, and can cause internal tissue damage to the samples, which has a significant impact on their edibility and marketability [8]. Therefore, there is an urgent need in the market for a non-destructive and efficient technique to assess the physicochemical indicators of lettuce.

Visible/near-infrared spectroscopy is a technique that utilizes the electromagnetic spectrum range of visible light and near-infrared light for analysis. It combines electronics, optics, information processing, and computer science techniques to integrate spectral information of samples with spatial information. By collecting the reflectance, absorption, and scattering spectral data of samples in the visible/near-infrared spectral range, visible/near-infrared spectroscopy can provide information about the external characteristics, internal physical structure, and chemical composition of the samples. In recent years, both domestic and international research has successfully utilized visible/near-infrared spectroscopy to predict soluble solids content, moisture content, vitamin C content, and other parameters in fruits and vegetables [9,10,11,12,13,14]. These research results demonstrate the enormous potential of visible/near-infrared spectroscopy in the quality assessment of fruits and vegetables.

In this study, lettuce was grown using hydroponic cultivation, and experiments were conducted to determine the SSC, moisture content, and vitamin C levels of lettuce at the rosette stage. The changes in the internal indicators of lettuce during the rosette stage were analyzed based on the number of growth days. Additionally, non-destructive testing of the physicochemical indicators of lettuce at the rosette stage was performed using visible/near-infrared spectroscopy. A portable device for visible/near-infrared spectroscopy was designed to achieve online non-destructive testing of the physicochemical indicators of lettuce at the rosette stage.

## 2. Materials and Methods

### 2.1. Experimental Materials

The growth cycle of lettuce generally consists of four stages: germination, seedling, rosette, and heading. The experimental samples in this study were the leaves of lettuce at the rosette stage. The experimental design was as follows: Firstly, a small amount of hot water at 55 °C was poured into clean glass bottles, and Italian Romaine lettuce seeds were soaked in the hot water for approximately 12 h. After soaking, the seeds were transferred to Petri dishes lined with filter paper, and the dishes were placed in a germination chamber (15 °C, no light) for 24 h. A foam board with 100 small holes measuring 1 cm × 1 cm each and a bottom board measuring approximately 2 m in length, 1 m in width, and 0.4 m in height were prepared. Nutrient solution was poured into the bottom board, and the roots of germinated seeds were inserted into the holes in the foam. The foam was placed in the nutrient solution in the bottom board, and the entire bottom board was placed in a growth chamber (temperature of 20 °C, humidity of 60% RH, and illumination of 30,000 Lx). During the growth period, nutrient solution (the variety of nutrient solution used was Hoagland’s Nutrient Solution, and the nutrient solution was configured in the ratio of 1g of powder to 1L of water) was added to the plants every 2 days, and the nutrient solution was replaced every week. When the plants reached the seedling stage, the foam with the seedlings was separated and transplanted until the plants reached the rosette stage. Then, lettuce plants at the rosette stage were selected, and intact and healthy leaves were harvested. The leaves were cut into small pieces measuring 25 mm × 25 mm and placed in labeled plastic bags for further analysis. Due to the small size of the cut leaves, each labeled leaf could only be used for the detection of one physicochemical indicator. The lettuce plants at the rosette stage are shown in Figure 1.

### 2.2. Measurement Indicators and Methods

#### 2.2.1. Determination of Soluble Solids Content

Lettuce soluble solids were measured using a Shen Guang WAY (2WAJ, Shanghai Li-Chen Bang Xi Instrument Technology Co., Ltd., (Shanghai, China)) Abel refractor device, using the following steps Place the leaves in a mortar and pestle to crush them into crumbs, strain the crumbs using filter paper, and drop the filtered juice onto the surface of the refracting prism using a dropper. Make sure the liquid is uniform, the field of view is intact, and there are no air bubbles. Next, open the visor, and close the reflector. Adjust the eyepiece view so that the crosshairs are clearly imaged. Make the demarcation line free of blue and orange by rotating and adjusting the handwheel so that the demarcation line is centered on the crosshairs. The value of the upper half of the reading was noted as the SSC content of the lettuce [15].

#### 2.2.2. Determination of Moisture Content

The moisture content of lettuce was measured using an electric blast drying oven. The electric blast drying oven was produced by Shanghai Li-Chen Bang Xi Instrument Technology Co., Ltd. with a power supply of 220 V/50 Hz and a blast power of 10 W. The drying oven was used to measure the moisture content of lettuce, selecting the same number of aluminum boxes as the total number of samples and weighing them to obtain the corresponding mass, $m_{0}$. The lettuce leaves were placed into the aluminum boxes and weighed to obtain the weight, $m_{1}$. The boxes were then placed into an electric hot air drying oven set at a temperature of 105 °C and dried for 6 h. After drying, the boxes were taken out and weighed. They were then placed back into the drying oven for an additional 15 min and weighed again. If there was a difference between the two weighing results, the aluminum boxes were placed back into the drying oven until the most recent two weighing results were the same. At this point, the weight, $m_{2}$, was recorded. The moisture content, m, was calculated using Equation (1).
(1)$$m=\frac{m_{1}-m_{2}}{m_{1}-m_{0}}$$

#### 2.2.3. Determination of Vitamin C

The vitamin C content of lettuce was measured using a vitamin C test kit, which was produced by Shanghai Yuan Ye Biologicals Co. (Shanghai, China). The fresh lettuce leaves were placed in a mortar and crushed. An accurate measurement of two grams of crushed fragments was taken and transferred into a 2 mL centrifuge tube. The reagents from the kit were prepared in the following order: tissue homogenate, $V_{C}$ buffer reagent, MPA working solution, and eight gradients (25, 50, 75, 100, 125, 150, 200, 250 µg/mL) of $V_{C}$ standard reagent. Using a clean dropper, the tissue homogenate was dropped into the centrifuge tube and thoroughly mixed until the liquid level reached 1.5 mL. The centrifuge tube containing the sample fragments was centrifuged for 5 min, and 0.1 mL of the supernatant was taken out and mixed with 0.1 mL of distilled water. After thorough mixing, the liquid representing the sample was obtained. Next, the sample liquid, the eight gradients of $V_{C}$ standard reagent, and distilled water were dropped into a 96-well plate. Three wells were used for the titration of each sample liquid. MPA working solution and $V_{C}$ buffer reagent were added in sequence and thoroughly mixed. The plate was placed in a water bath at 30 degrees Celsius and allowed to react for 20 min. Before starting the measurements, the spectrophotometer was zeroed with a blank. Then, the plate was placed in the spectrophotometer, and the parameters were set to 760 nanometers. The absorbance results of each well were recorded.

The absorbance values corresponding to the eight gradients of $V_{C}$ standard reagent were plotted on the *y*-axis, while the concentrations of the $V_{C}$ standard reagent were plotted on the *x*-axis. This generated a standard curve for $V_{C}$. A regression equation was obtained from the standard curve. By substituting the absorbance values of the sample determination wells into the regression equation, the vitamin C content of the sample could be calculated.

### 2.3. Visible/Near-Infrared Spectral Acquisition

The visible/near-infrared diffuse reflectance spectroscopy acquisition system mainly consists of an adjustable halogen lamp light source, glass fiber optic cable, Maya2000Pro spectrometer, and 84UV collimating lens, as shown in Figure 2.

The visible/near-infrared spectrometer used in this experiment was the Maya2000Pro fiber optic spectrometer (Ocean Optics, Orlando, FL, USA). It is a compact and high-performance spectrometer that can measure spectra in the visible to near-infrared range. It provides high resolution and high sensitivity for spectral measurements and is widely used for absorbance, raw light intensity, reflectance, and transmittance measurements of solid and liquid samples [16,17]. In order to obtain stable and reliable spectral information, a powerful and stable light source, the WC-150 (Shanghai Winci Optoelectronics Technology Co., Ltd., China, Shanghai, China) 150 W halogen lamp cold light source, was chosen for this experiment. This light source has adjustable light intensity and is equipped with a cooling device. Its effective wavelength range is from 400 nm to 900 nm. When combined with the spectrometer, it can provide spectra in the visible/near-infrared range from 400 nm to 1000 nm.

Before collecting the visible/near-infrared spectra of the lettuce samples, it was necessary to acquire the dark spectrum and reference spectrum, using the following method. After preheating the instrument for 30 min, turn on the power switch, and adjust the brightness to 0 to collect the dark spectrum and save it. Then, adjust the light source to the maximum, and place a white standard board to collect the reference spectrum and save it. Set the acquisition mode of the spectrometer to “reflection”, and adjust the light source intensity to 50%. Place the lettuce sample on the sample stage, ensuring that the center of the sample aligns with the collimating lens on a vertical line. Collect the diffuse reflectance spectra data of the sample.

### 2.4. Modeling and Evaluation Methodology

The PLSR (Partial Least Squares Regression), LSSVMs (Least Squares Support Vector Machines), and MLR (Multiple Linear Regression) models were established based on the spectral information of lettuce, including the original spectra and the spectra after feature wavelength extraction. These models were used to predict the SSC (soluble solids content), moisture content, and vitamin C value of lettuce.

In order to perform feature wavelength extraction and model building, this study utilized three methods: CARS (Competitive Adaptive Reweighted Sampling), SPA (Successive Projection Algorithm), and UVE (Uninformative Variable Elimination) to select feature variables with relevant information. These selected feature variables were used to establish three models: Partial Least Squares Regression (PLSR), Least Squares Support Vector Machines (LSSVMs), and Multiple Linear Regression (MLR). MLR is a statistical method based on linear models, also known as Inverse Least Squares, which is widely used in statistics and data analysis. It is particularly suitable for exploring the linear relationship between spectral data and physicochemical properties. PLSR is widely used for information extraction and chemical feature analysis. It combines the advantages of Multiple Linear Regression, principal component analysis, and correlation analysis and can effectively handle high-dimensional data and multicollinearity issues. LSSVM is based on the Support Vector Machine approach and solves the multivariate linear equation system instead of quadratic programming problems. By comparing and analyzing these three models, the performance differences in feature wavelength extraction and modeling can be evaluated.

The accuracy and stability of the models are evaluated based on the correlation coefficient of calibration ($R_{C}$) and root mean square error of calibration (*RMSEC*) for the training set, as well as the correlation coefficient of prediction ($R_{P}$), root mean square error of prediction (*RMSEP*), and residual predictive deviation (*RPD*). A higher value of $R_{C}$ and $R_{P}$, closer to 1, indicates the higher stability and predictive accuracy of the model. A lower value of *RMSEC* and *RMSEP*, closer to 0, also indicates better model performance [18]. The *RPD* value is used to assess the model’s predictive ability. When the *RPD* value is less than 1.5, it suggests poor prediction capability and insufficient stability. When the *RPD* value is between 1.5 and 2.0, it indicates good stability, allowing for rough quantitative predictions. When the *RPD* value is greater than or equal to 2.0, the model exhibits excellent stability and applicability, enabling accurate qualitative predictions [19].
(2)$$R_{c}=\sqrt{\sum_{i=1}^{k_{c}}{\left(y_{mi}-y_{ni}\right)}^{2}}/\sqrt{\sum_{i=1}^{k_{c}}{\left(y_{mi}-y_{j}\right)}^{2}}$$
(3)$$R_{p}=\sqrt{\sum_{i=1}^{k_{p}}{\left(y_{mi}-y_{ni}\right)}^{2}}/\sqrt{\sum_{i=1}^{k_{c}}{\left(y_{mi}-y_{j}\right)}^{2}}$$
(4)$$RMSEC=\sqrt{\frac{1}{n_{c}}\sum_{i=1}^{k_{c}}{\left(y_{mi}-y_{ni}\right)}^{2}}$$
(5)$$RMSEP=\sqrt{\frac{1}{n_{p}}\sum_{i=1}^{k_{p}}{\left(y_{mi}-y_{ni}\right)}^{2}}$$
(6)$$RPD={std}_{p}/RMSEP$$

In the equation, $y_{mi}$ represents the predicted value of the i-th sample in the calibration set or prediction set, $y_{ni}$ represents the actual measured value of the i-th sample in the calibration set or prediction set, $y_{j}$ represents the average value of all actual measured values of the calibration set samples or prediction set samples, and *k* represents the total number of samples involved in the calibration set or prediction set. ${std}_{p}$ represents the standard deviation of the prediction set.

## 3. Results

### 3.1. Changes in Physicochemical Index Parameters with Time in Lettuce at Rosette Stage

The lettuce plants are selected when they reach the heading stage for the experiments. To prevent interference among plants due to limited space, the plants are divided into individual seedlings. On the 5th day of growth, the first experiment is conducted. The lettuce plants are harvested, and intact and clean leaves are selected. The leaves are then cut into small pieces measuring 25 mm × 25 mm. For each experiment, a total of 90 leaf pieces are cut and divided equally into 3 portions for the detection of SSC, moisture content, and vitamin C. The experiment cycle is set to 30 days, allowing a total of 6 experiments to be conducted. This results in obtaining 180 sets of data for each physicochemical index. The average values of the lettuce physicochemical indices are plotted against the number of days of growth in the heading stage, as shown in Figure 3.

According to Figure 3, the relative content of SSC in lettuce shows an increasing trend before the 10th day of the heading stage, rising from 2.51 to 2.66. Between 10 and 15 days of growth, the SSC content gradually decreases, reaching a minimum of 2.26. After 15 days, the SSC content starts to increase again, reaching its maximum value of 3.23 on the 30th day. The moisture content of lettuce shows a decreasing trend before the 20th day of the heading stage, decreasing from 95.48 to 90.65. Between 20 and 30 days of growth, the moisture content slightly increases, reaching 90.99 on the 30th day. The relative content of vitamin C in lettuce gradually decreases between the 10th day of the heading stage, declining from 16.93 to 10.83. Between 10 and 15 days of growth, the relative content of vitamin C slightly decreases, reaching 10.64 on the 15th day. Between 15 and 20 days of growth, the relative content of vitamin C steeply increases, reaching 28.18 on the 20th day. Between 20 and 30 days of growth, the relative content of vitamin C initially decreases from 28.18 to 22.07, and then increases to 24.31.

Overall, during the heading stage, the SSC of lettuce shows a general trend of increasing, then decreasing, and gradually increasing again. The moisture content tends to gradually decrease, followed by a slight increase. The relative content of vitamin C shows a pattern of decreasing, followed by a sharp increase, then decreasing again, and finally increasing.

### 3.2. Characteristics of Visible/Near-Infrared Diffuse Reflectance Spectra of Lettuce at Rosette Stage and Changing Law

#### 3.2.1. Characterization of Visible/Near-Infrared Diffuse Reflectance Spectra of Lettuce at Rosette Stage

Figure 4 shows the diffuse reflectance spectroscopy of lettuce in the range of 400 nm to 1000 nm. Based on the curves in the figure, it can be observed that there are distinct peaks and valleys in the reflectance curves at 449 nm, 559 nm, 670 nm, 717 nm, 812 nm, and 901 nm. The wavelength range of 400 nm to 780 nm corresponds to the visible light spectrum, which reflects the color and luminous information of lettuce. Among them, 449 nm is in the violet light range, while 559 nm is in the green light range. Due to the weak absorption of green light by chlorophyll in the leaf, there is a pronounced reflection peak around 559 nm, which is the main reason for the green color of the leaf. Around 670 nm, the leaf undergoes photosynthesis, leading to an enhanced absorption of red light by chlorophyll. Therefore, there is a distinct absorption valley in that wavelength range. In the range of 670 nm to 720 nm, the decrease in chlorophyll content leads to a decrease in pigment content, resulting in an increase in reflectance [20]. The peaks and valleys in the reflectance spectrum of lettuce leaves in the range of 800 nm to 1000 nm are related to the vibrations of intermolecular chemical bonds. At 901 nm, there is a pronounced peak due to the vibrations of hydroxyl (O-H) and amino (N-H) groups in the leaf [21].

#### 3.2.2. Changing Pattern of Visible/Near-Infrared Diffuse Reflectance Spectra of Lettuce at Rosette Stage

Figure 5 shows the average spectrum of the 90 lettuce leaf samples collected in each experiment. The spectrum data were obtained in the wavelength range of 400 nm to 1000 nm, with a total of 1360 wavelength points.

From the graph, it can be observed that the reflectance of the visible/near-infrared spectrum of lettuce during the heading stage shows a consistent trend of variation. As the number of days of growth increases, the overall trend of reflectance is gradually increasing. From the 5th day to the 10th day, the reflectance at the peaks of 559 nm and 901 nm, as well as the valley at 812 nm, shows a slow increase, while the reflectance at the peak of 717 nm shows no significant change. From the 10th day to the 15th day, there is a significant increase in reflectance at the peaks of 559 nm, 717 nm, and 901 nm, as well as the valley at 818 nm. From the 15th day to the 20th day, the reflectance at the peak of 559 nm shows an increasing trend, while the reflectance at the peaks of 717 nm and 901 nm slightly decreases, and the reflectance at the valley of 812 nm shows no significant change. From the 20th day to the 30th day, there is a sharp increase followed by a gradual increase in reflectance at the peaks of 559 nm, 717 nm, and 901 nm, as well as the valley at 818 nm. There is a noticeable difference in the reflectance spectra between the 5th day and the 30th day of the heading stage of lettuce.

### 3.3. Classification of the Sample Set

In this study, the Kennard–Stone (KS) method was chosen to divide the sample set [22]. There were 180 samples of lettuce for SSC, moisture content (MC), and vitamin C (VC), which were divided in a ratio of 3:1. The calibration set consisted of 135 samples, while the prediction set had 45 samples. The statistical results of the samples are shown in Table 1. From the observation of Table 1, the SSC content of lettuce leaves ranged from 1.48°Brix to 4.84°Brix, the moisture content ranged from 87.89% to 97.27%, and the vitamin C values ranged from 6.33% to 44.63%. After using the KS method for division, the sample indicators in the calibration set had a wider range than the prediction set, ensuring that the data in the prediction set could be fully incorporated into the model established by the calibration set.

### 3.4. Spectral Data Preprocessing

Preprocessing the raw spectra can effectively remove noise from the spectral curves. Common preprocessing methods include mean variance normalization (MVN), centering (Center), normalization (Nor), moving average method (MA), Savitzky–Golay convolution smoothing (S_G), standard normal variate (SNV) transformation, and multiplicative scatter correction (MSC). By comparing their effects in the Partial Least Squares Regression (PLSR) model, the preprocessing method that best matches the three indicators can be identified [23,24,25,26,27].

From Table 2, it can be observed that for the SSC indicator, using the S_G smoothing preprocessing method significantly improves the performance of the PLSR model. The models built with other preprocessing methods show similar performance to the model built with the raw spectra. For the moisture content indicator, the models built with the Center, Nor, S_G, and MSC preprocessing methods all show significant improvement. Among them, the Nor method has the best overall performance. For the vitamin C indicator, the S_G preprocessing method yields the best results, while the other methods do not show a significant improvement compared to the raw spectra.

Overall, the optimal spectral preprocessing method for the SSC and vitamin C indicators is S_G, while the optimal spectral preprocessing method for the moisture content indicator is Nor. The original spectra and the spectral curves after different spectral pre-processing are shown in Figure 6.

### 3.5. Spectral Characteristic Wavelength Selection

The original spectra of the collected lettuce contain a large number of wavelength points, among which there may be many features that are unrelated to the indicators. Therefore, it is necessary to perform feature extraction in the full wavelength range to extract the wavelengths that are relevant to the corresponding indicators.

#### 3.5.1. CARS Method for Extracting Characteristic Wavelengths

Taking the moisture content of lettuce leaves as an example, this study set the number of principal components to 11, performed 160 resampling iterations with a resampling rate of 0.8, and used 5–fold cross–validation for calculation. The results are shown in Figure 7. Figure 7a displays the variation in the selected number of feature wavelengths, which gradually slows down as the number of iterations increases. Figure 7b shows the change in root mean square error of cross–validation (RMSECV) with an increasing number of iterations. At the beginning stage, ineffective feature data in the spectra are removed, resulting in a gradual decrease in RMSECV. However, as the number of iterations increases, effective feature data are also eliminated, leading to a gradual increase in RMSECV. Figure 7c illustrates that the minimum point of RMSECV corresponds to the vertical midline position of the regression coefficients, occurring at 92 iterations, with 33 selected feature wavelengths. At this point, RMSECV reaches its minimum value, indicating the best predictive performance of the model. Therefore, the optimal subset of feature variables can be determined as the spectral wavelength variable set corresponding to the minimum RMSECV, which includes 33 feature wavelengths.

#### 3.5.2. UVE Method for Extracting Characteristic Wavelengths

Taking the moisture content of lettuce leaves as an example, this study set the threshold to 0.9 and the number of principal components to 13. The results are shown in Figure 8. The yellow curve on the left represents the stable values of the moisture content spectral variable matrix, while the red curve on the right represents the spectral random noise variable matrix. The two horizontal blue dashed lines represent the maximum threshold for random noise (±3.72). The variables between the dashed lines are eliminated as spectral information, while the variables outside the dashed lines are selected as feature variable information. By employing the Uninformative Variable Elimination (UVE) method, a total of 445 feature wavelengths are selected, accounting for 32.72% of the original spectra.

#### 3.5.3. SPA Method for Extracting Characteristic Wavelengths

Taking the moisture content of lettuce leaves as an example, the process of extracting feature wavelengths using the Sequential Projection Algorithm (SPA) is shown in Figure 9. The SPA algorithm selects feature variables based on the trend of the root mean square error (RMSE). From Figure 9a, it can be observed that as the number of variables increases, the RMSE initially decreases rapidly, indicating the elimination of irrelevant information in the spectra. Afterward, the RMSE tends to flatten, indicating the elimination of irrelevant variables. The turning point in the curve is chosen as the selected feature variables. It is important to note that the turning point is not necessarily the lowest point of the curve but rather the point at which the RMSE reaches a reasonable range. At this point, the number of wavelength points is 16, and the value of RMSE is 1.27637. The selected feature wavelength variables account for 1.18% of the full spectrum. The indices of the selected wavelength points are shown in Figure 9b.

### 3.6. Modeling and Validation of Effects

For the soluble solids content and vitamin C indicators, the optimal preprocessing method is S_G, while for the moisture content indicator, the optimal preprocessing method is Nor. The spectral data of the three indicators were preprocessed using the respective optimal methods. The preprocessed data were then subjected to feature wavelength extraction, with the selected feature wavelengths serving as inputs for the MLR, PLSR, and LSSVM models. The experimental results of the three indicators of lettuce leaves were used as the outputs of the models.

#### 3.6.1. Modeling

For the soluble solids content (SSC) indicator, the combination of the S_G preprocessing method and UVE feature wavelength extraction yielded better results for the MLR, PLSR, and LSSVM models, the results are shown in Table 3, Table 4 and Table 5. Among them, the PLSR model outperformed the MLR and LSSVM models. The PLSR model achieved a calibration set $R_{C}$ of 0.8946 and a prediction set $R_{P}$ of 0.8648, with a *PRD* value of 1.9685.

For the moisture content indicator, the combination of the Nor preprocessing method with CARS feature wavelength extraction yielded better results for the MLR and PLSR models. However, for the LSSVM model, the UVE feature wavelength extraction method performed the best. Among the models, the PLSR model achieved the best prediction for the moisture content indicator. The PLSR model had a calibration set $R_{C}$ of 0.8895 and a prediction set $R_{P}$ of 0.8578, with a *PRD* value of 1.9568.

For the vitamin C indicator, the combination of the S_G preprocessing method with UVE feature wavelength selection method performed best on the MLR, PLSR, and LSSVM models. Other feature wavelength selection methods resulted in decreased performance. Among the models, the PLSR model outperformed the MLR and LSSVM models. The PLSR model had a calibration set $R_{C}$ of 0.8345 and a prediction set $R_{P}$ of 0.8047, with a *PRD* value of 1.6689.

After comprehensive comparison, it can be concluded that the PLSR model performs better than the MLR and LSSVM models for predicting the physicochemical indicators (SSC, moisture content, and vitamin C) of lettuce during the head-forming stage. The optimal feature wavelength selection method for both the SSC and vitamin C indicators is UVE, while the optimal feature wavelength selection method for the moisture content indicator is CARS. Both the MLR and PLSR models are linear regression models, but the PLSR model consistently outperforms the MLR model for predicting the physicochemical indicators of lettuce during the head-forming stage. Therefore, the PLSR model is more suitable for predicting the content of physicochemical indicators in lettuce during the head-forming stage. The index plots of the characteristic wavelengths selected for the three metrics under the optimal model are shown in Figure 10.

#### 3.6.2. Validation of Effects

We established PLSR models for the prediction of soluble solids content, moisture content, and vitamin C content in lettuce leaves during the head-forming stage using the optimal spectral preprocessing and feature wavelength selection methods. The predicted values of the training set and prediction set samples, as well as the actual measurements of the total samples, are shown in Figure 11.

### 3.7. Portable Visible/Near-Infrared Spectroscopy Testing Device for Lettuce Physical and Chemical Indicators

#### 3.7.1. Detection Device Design

The developed visible/near-infrared spectroscopy acquisition device consists of a Maya2000Pro spectrometer, a 50 W halogen lamp, a collimating lens, and a glass fiber, as shown in Figure 12. The overall dimensions of the detection device are 17 cm × 26 cm × 16 cm. The Maya2000Pro spectrometer is easy to operate and has high sensitivity and low noise, making it suitable for integration with a small computer. Choosing a suitable illumination source can meet the requirements of sample spectral data acquisition and improve the accuracy of predicting the physicochemical indicators of the samples. The device uses an adjustable 50 W halogen lamp as the illumination source. Halogen lamps are cost-effective, have a long lifespan, and offer stable light output, making them suitable for sample spectral acquisition. Two 50 W halogen lamps are symmetrically arranged on the inner wall of the device. The collimating lens is positioned at the top of the device, with its center aligned with the center of the sample holder on the same vertical line. The glass fiber is connected to the collimating lens using an internal threaded tube, and the glass fiber is then connected to the Maya2000Pro spectrometer to collect the reflected light from the samples for visible/near-infrared spectroscopy. The control component used in the device is a Raspberry Pi 4B, which is a powerful and widely used single-board computer. It features a 1.5 GHz quad-core ARM Cortex-A72 processor, 4 GB of memory, and a rich set of GPIO pins. The Raspberry Pi 4B is connected to the 50 W halogen lamp through the GPIO pins for digital input and output. It is also connected to the Maya2000Pro spectrometer via a USB interface, enabling the control of the spectrometer and adjustment of the halogen lamp’s intensity.

#### 3.7.2. Software Design

In the MATLAB 2021a GUI, there is a rich library of components that can meet the requirements for software design. To use MATLAB 2021a on a Raspberry Pi, you need to first download the MATLAB 2021a hardware support package to the SD card of the Raspberry Pi and perform the necessary parameter settings. The designed software interface is shown in Figure 13. The software reads the acquired lettuce spectral information and provides two preprocessing methods (S_G, Nor) and two feature wavelength selection methods (CARS, UVE) to process the collected spectral data. The preprocessed spectral curves and the selected wavelength bands for feature extraction are displayed on the corresponding interface. By default, the PLSR model is used to predict the content of physicochemical indicators. Clicking on the “Indicators predicted” button will display the predicted content of the physicochemical indicators of lettuce on the interface. Finally, the predicted data can be saved, and clicking on the “Exit Software” button will exit the software.

### 3.8. Device Performance Testing Experiment

After completing the construction of the portable device for lettuce during the head-forming stage, it is necessary to test and verify its performance. The experimental material used is lettuce leaves during the head-forming stage, and the sample preparation is consistent with Section 2.1. Performance testing of the device involves comparing the predicted values of the physicochemical indicators with the actual values. For each growth cycle of lettuce during the head-forming stage, 30 experimental samples are tested every 5 days. These 30 samples are evenly distributed for the detection of the content of the three physicochemical indicators. The total number of experimental samples is 180, with 60 samples for each indicator. The results of the device performance testing are shown in Figure 14.

According to Figure 14, the correlation coefficients for the prediction of SSC, moisture content, and vitamin C indicators in the prediction set of lettuce during the head-forming stage are 0.8215, 0.8472, and 0.7671, respectively. The root mean square errors of prediction (*RMSEP*) for the prediction set are 0.5348, 1.5813, and 2.3347, respectively. The prediction results for the physicochemical indicators are slightly lower compared to the results in Section 3.6, but the difference is not significant. This could be due to the influence of external environmental factors during the spectral acquisition process. The test results indicate that the device provides results for the physicochemical indicators of lettuce during the head-forming stage that are close to the actual measurements. The test results meet the requirements for practical detection.

## 4. Discussion

This study focused on hydroponic lettuce during the head-forming stage. A visible/near-infrared spectroscopy acquisition device was used to obtain the diffuse reflectance spectra of lettuce. By combining the actual measurements of the physicochemical indicators of lettuce, a non-destructive detection model for the physicochemical indicators of lettuce during the head-forming stage was developed based on visible/near-infrared spectroscopy technology. Based on the optimal model results, a portable visible/near-infrared spectroscopy detection device for lettuce during the head-forming stage was designed. Based on the completed experimental results, the following research conclusions can be drawn:

Research on the variation in internal physicochemical indicators and spectral characteristics of lettuce during the head-forming stage. Visible/near-infrared spectroscopy technology was used to investigate the changes in the internal physicochemical indicators (such as SSC, moisture content, and vitamin C) of hydroponic lettuce during the head-forming stage with increasing growth days. The results showed that with the increase in head-forming days, the relative content of SSC initially increased, then decreased, and gradually increased again. The moisture content initially decreased and then slightly increased. The relative content of vitamin C decreased initially, sharply increased, then decreased again, and finally increased. Based on the visible/near-infrared spectroscopy curves of lettuce during the head-forming stage, it was observed that at 449 nm, 559 nm, 670 nm, 717 nm, 812 nm, and 901 nm, the spectral reflectance curves exhibited distinct peaks and valleys. Throughout the entire growth period of lettuce during the head-forming stage, the reflectance variation trend of the visible/near-infrared spectroscopy curves remained consistent. With the increase in growth days, the overall reflectance of the spectral curves showed a gradual increase. There were significant differences in the reflectance spectral curves between the early and late stages of head-forming.

Research on the detection model of physicochemical indicators of lettuce during the head-forming stage based on visible/near-infrared spectroscopy technology. Hydroponic lettuce during the head-forming stage was used as the research object. The SSC, moisture content, and vitamin C indicators, as well as the visible/near-infrared diffuse reflectance spectra of lettuce, were collected. The KS algorithm was used to divide the sample set. By comparing the effects of seven spectral preprocessing methods (MVN, Center, Nor, MA, S_G, SNV, MSC), three spectral feature wavelength selection methods (CARS, UVE, SPA), and three models (MLR, PLSR, LSSVM), the optimal model structures corresponding to the three indicators were identified. The results showed that the optimal model structure for SSC and vitamin C indicators of lettuce during the head-forming stage was S_G-UVE-PLSR, while the optimal model structure for the moisture content indicator was Nor-CARS-PLSR. The correlation coefficients of PLSR models for SSC, moisture content, and vitamin C in the calibration set were 0.8946, 0.8895, and 0.8345, respectively and, in the prediction set, were 0.8648, 0.8578, and 0.8047, respectively. The residual prediction deviations of the prediction set were 1.9685, 1.9568, and 1.6689, respectively. The results based on the optimal model structure demonstrated the feasibility of using visible/near-infrared spectroscopy technology for the detection of the internal physicochemical indicators of lettuce during the head-forming stage, providing data validation and technical support for the development of portable devices.

Development of a portable visible/near-infrared spectroscopy detection device for physicochemical indicators of lettuce during the head-forming stage. A portable detection device for the physicochemical indicators of lettuce during the head-forming stage was developed, incorporating the optimal model. This device enables the prediction of physicochemical indicator parameters of lettuce during the head-forming stage. The results showed that the correlation coefficients of the prediction set for SSC, moisture content, and vitamin C indicators of 180 lettuce samples were 0.8215, 0.8472, and 0.7671, respectively. The root mean square errors (*RMSEP*) of the prediction set were 0.5348, 1.5813, and 2.3347, respectively. The predicted values of the physicochemical indicators showed small differences from the true values, indicating that the obtained results met the basic accuracy requirements. The detection device for physicochemical indicators of lettuce during the head-forming stage can simultaneously measure SSC, moisture content, and vitamin C in lettuce.

## Figures and Tables

**Figure 1 foods-13-01863-f001:**
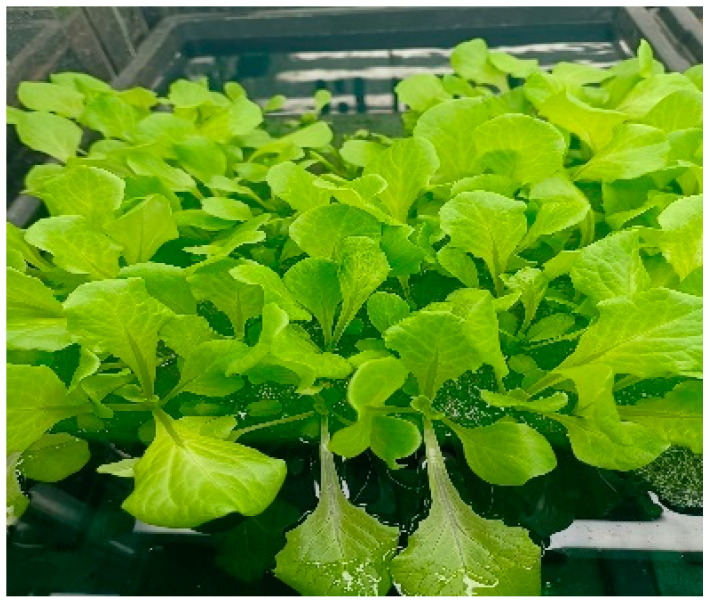
Lettuce plants at rosette stage.

**Figure 2 foods-13-01863-f002:**
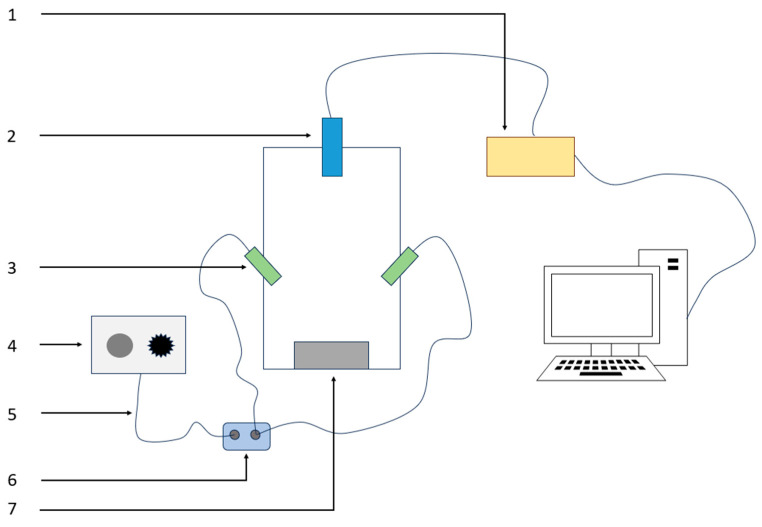
Visible/near-infrared Spectrum Acquisition Unit. 1. Maya2000Pro spectrometer; 2. collimating lens; 3. light source; 4. halogen light source; 5. optical fiber; 6. splitter; 7. carrier stage.

**Figure 3 foods-13-01863-f003:**
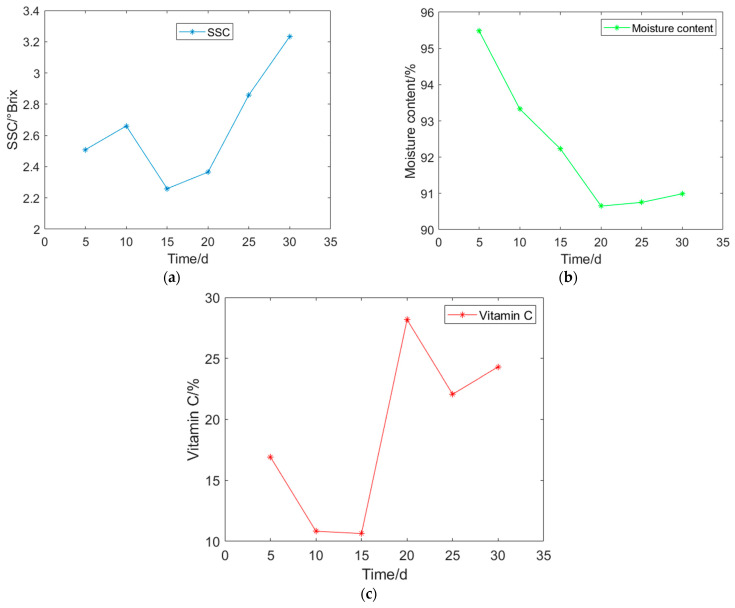
Changing trend of physicochemical indexes of lettuce at rosette stage: (**a**) SSC changing trend; (**b**) Moisture content changing trend; (**c**) Vitamin C changing trend.

**Figure 4 foods-13-01863-f004:**
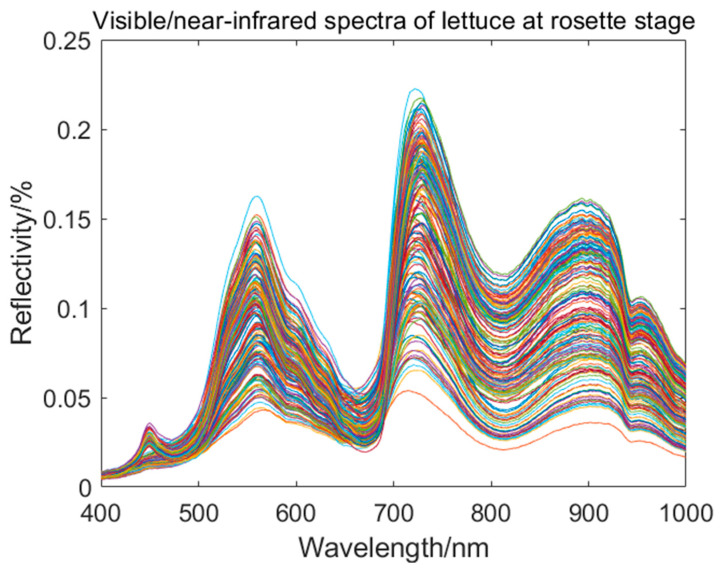
Visible/near-infrared spectral curves of lettuce at rosette stage.

**Figure 5 foods-13-01863-f005:**
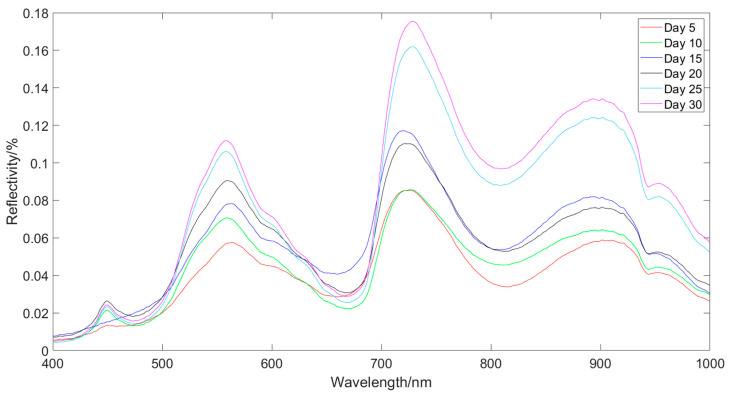
Average spectral curve of lettuce at rosette stage.

**Figure 6 foods-13-01863-f006:**
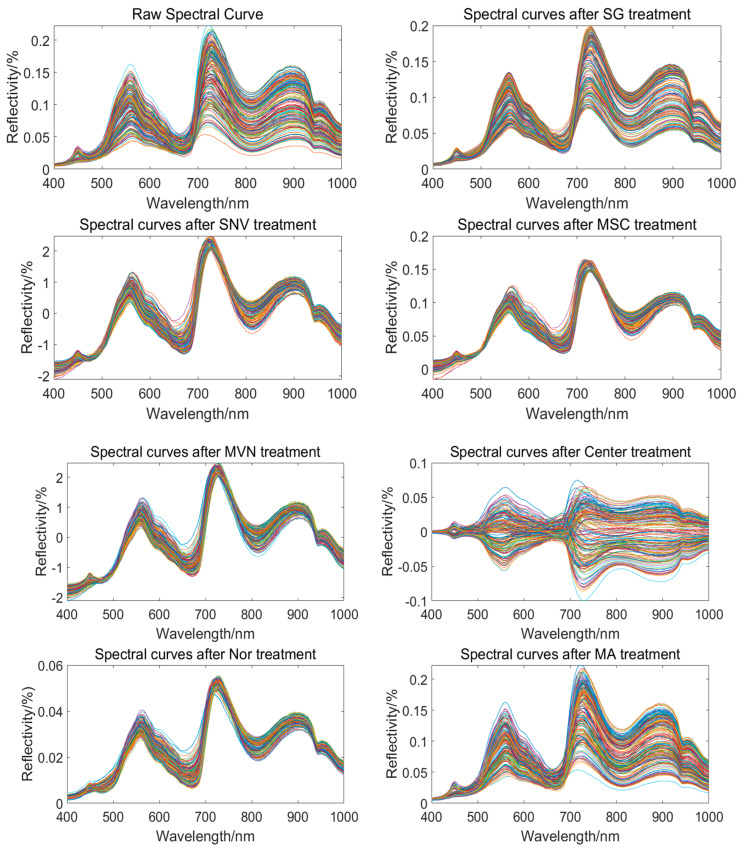
Raw spectra and full–wavelength spectral profiles of different preprocessing methods.

**Figure 7 foods-13-01863-f007:**
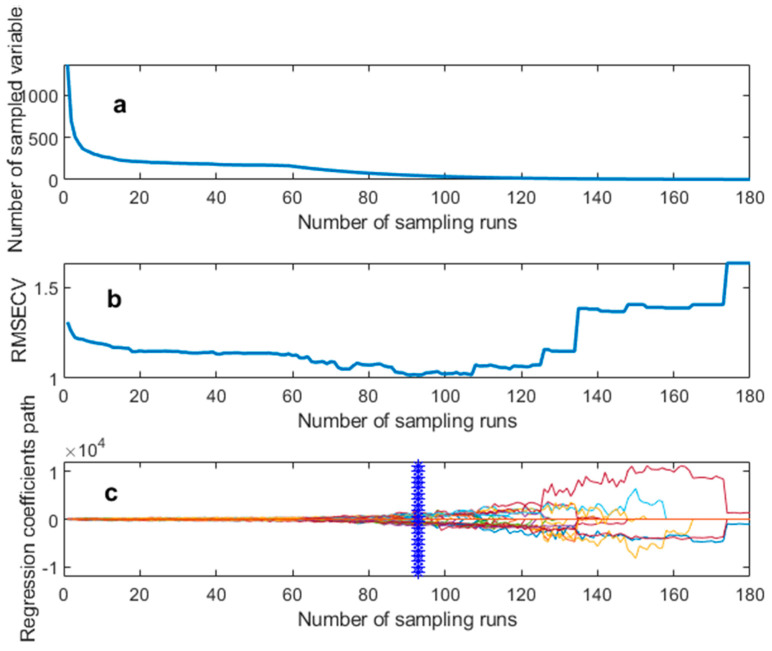
Process of selecting the characteristic wavelength for lettuce leaves’ moisture content using the CARS algorithm: (**a**) Number of spectral characterization variables; (**b**) RMSECV changing trend; (**c**) Regression coefficient path.

**Figure 8 foods-13-01863-f008:**
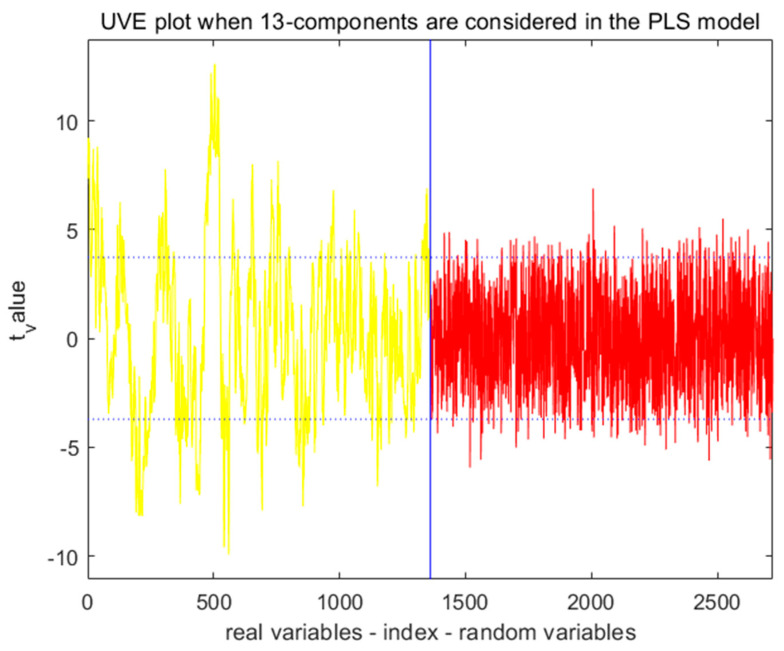
Process of selecting the characteristic wavelength for lettuce leaves’ moisture content using the UVE algorithm.

**Figure 9 foods-13-01863-f009:**
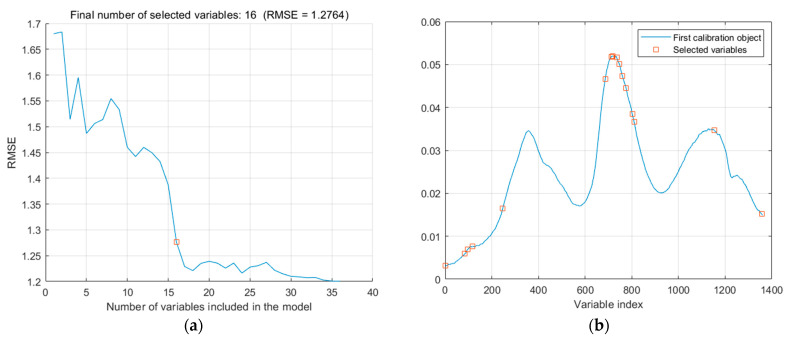
Process of selecting the characteristic wavelength for lettuce leaves’ moisture content using the SPA algorithm: (**a**) RMSE changing trend; (**b**) Selected characteristic wavelength points.

**Figure 10 foods-13-01863-f010:**
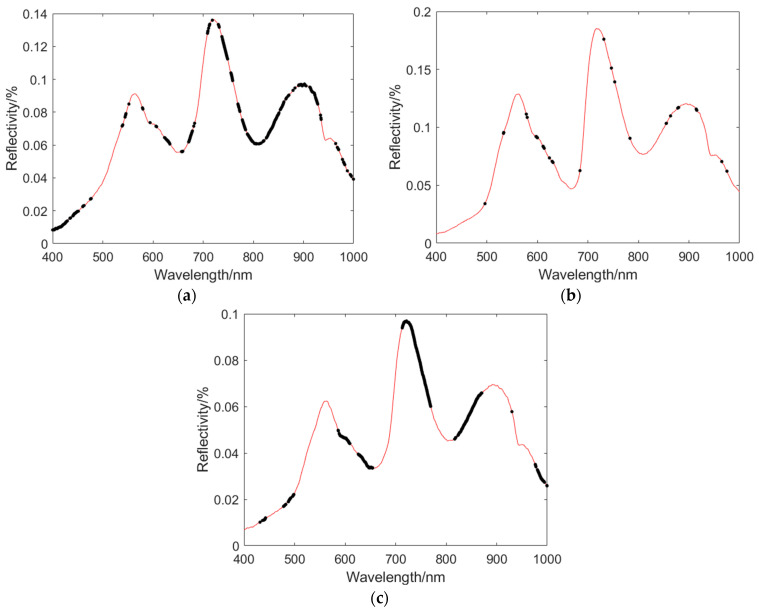
Optimal model feature wavelength point index: (**a**) Index of characteristic wavelength points of the SSC indicator; (**b**) Index of characteristic wavelength points of the moisture content indicator; (**c**) Index of characteristic wavelength points of the vitamin C indicator.

**Figure 11 foods-13-01863-f011:**
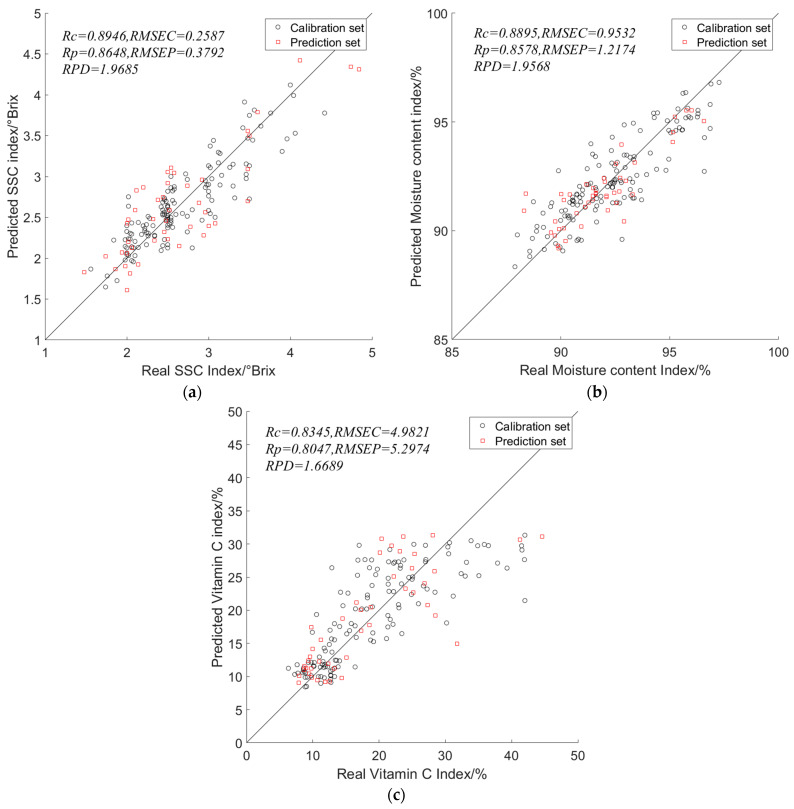
Optimal model prediction results of physicochemical indexes in lettuce at rosette stage: (**a**) Optimal model prediction results for SSC indicators; (**b**) Optimal model prediction results for moisture content indicators; (**c**) Optimal model prediction results for vitamin C indicators.

**Figure 12 foods-13-01863-f012:**
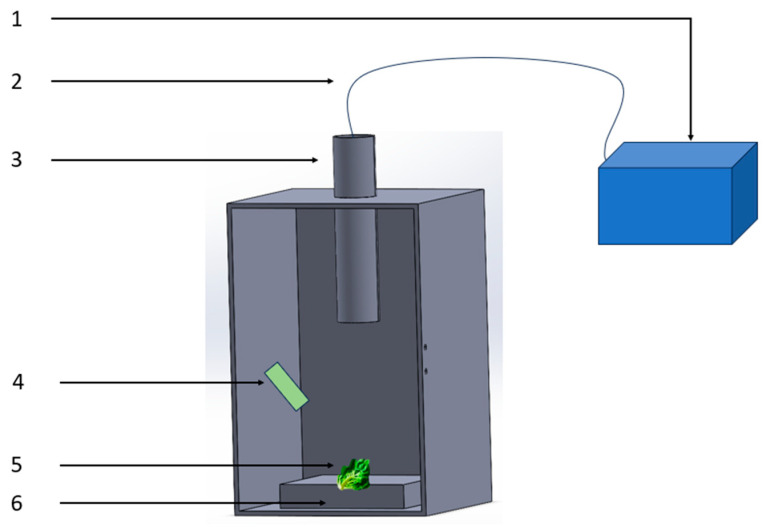
Detection device design. 1. Maya2000Pro spectrometer; 2. glass optical fiber; 3. collimating lens; 4. halogen lamp light source; 5. lettuce sample; 6. carrier table.

**Figure 13 foods-13-01863-f013:**
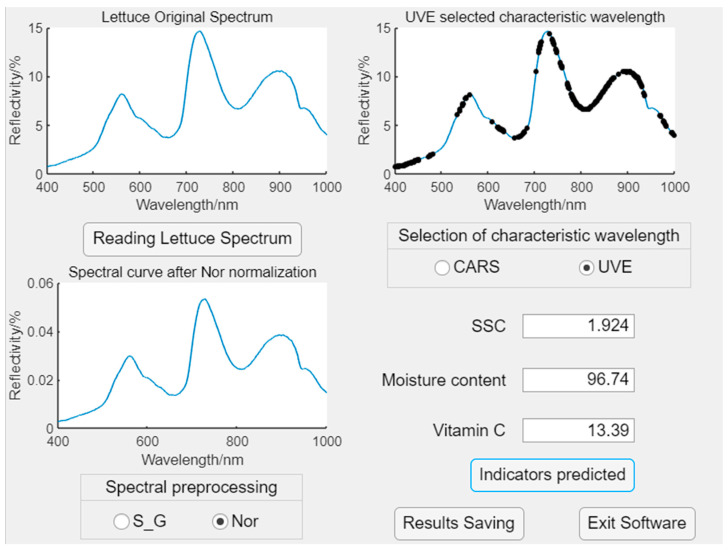
Real-time analysis software design.

**Figure 14 foods-13-01863-f014:**
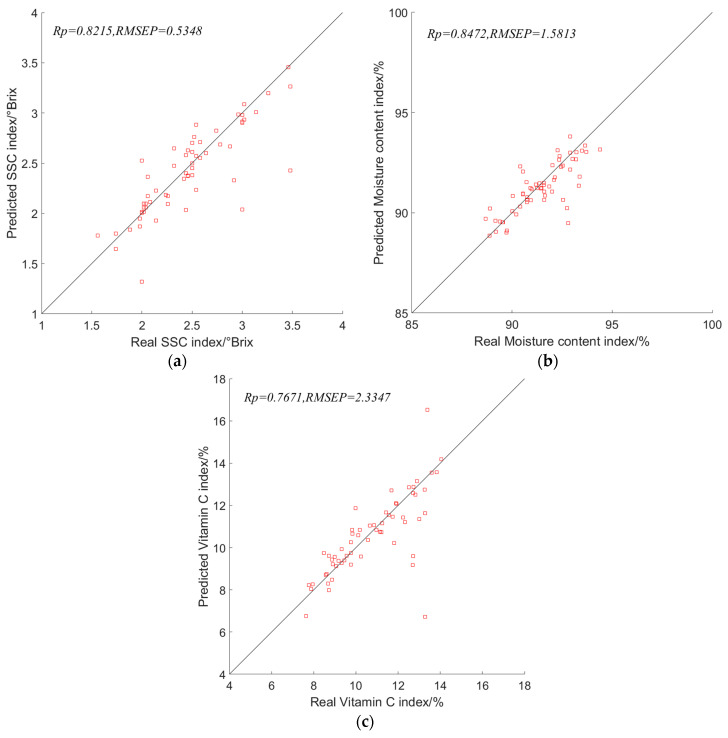
Validation results of portable rosette stage lettuce physicochemical index testing device: (**a**) Predicted results of SSC indicators; (**b**) Predicted results of moisture content indicators; (**c**) Predicted results of vitamin C indicators.

**Table 1 foods-13-01863-t001:** Results of KS algorithm to divide the sample set.

Indicators	Sample Set	Maximum Values	Minimum Value	Average Value	Standard Deviation
SSC/°Brix	calibration set	4.84	1.48	2.68	0.6323
	prediction set	4.00	1.74	2.56	0.5422
MC/%	calibration set	97.27	88.57	92.31	2.0924
	prediction set	96.59	87.89	92.03	2.3821
VC/%	calibration set	44.63	6.33	18.74	8.9190
	prediction set	42.00	7.28	19.09	8.8027

**Table 2 foods-13-01863-t002:** Raw spectra and PLSR modeling for seven preprocessing methods.

Indicators	Preprocessing Methods	Lvs Number of Principal Factors	Calibration Set		Prediction Set		*RPD*
			$R_{C}$	*RMSEC*	$R_{P}$	*RMSEP*	
SSC	RAW	16	0.8758	0.3041	0.7093	0.3800	1.4268
	MVN	13	0.8462	0.3357	0.6640	0.4062	1.3348
	Center	16	0.8761	0.3034	0.7087	0.3824	1.4247
	Nor	16	0.8711	0.3093	0.6878	0.3944	1.3748
	MA	16	0.8508	0.3310	0.7069	0.3808	1.4238
	S_G	21	0.8771	0.2718	0.8508	0.3839	1.9130
	SNV	14	0.8602	0.3222	0.6746	0.4024	1.3460
	MSC	11	0.8047	0.3658	0.7749	0.3953	1.4640
MC	RAW	15	0.5730	2.8119	0.3948	5.2439	0.4543
	MVN	15	0.9749	0.4649	0.7413	1.6321	1.4596
	Center	13	0.9587	0.5824	0.8374	1.2601	1.8749
	Nor	14	0.9610	0.5767	0.8485	1.2654	1.8824
	MA	16	0.5661	2.7600	0.4105	5.2676	0.4522
	S_G	16	0.8493	1.1696	0.8465	1.0296	1.8947
	SNV	15	0.9745	0.4972	0.7801	1.2984	1.5405
	MSC	14	0.9786	0.4455	0.8391	1.1571	1.8421
VC	RAW	13	0.9041	3.7928	0.6560	6.7292	1.3124
	MVN	12	0.8973	3.9184	0.6534	6.6583	1.3265
	Center	13	0.9244	3.5724	0.6604	6.6912	1.3197
	Nor	13	0.9080	3.7184	0.6668	6.5634	1.3457
	MA	13	0.8749	4.2981	0.6636	6.6278	1.3326
	S_G	13	0.7923	5.3987	0.7888	5.4520	1.6285
	SNV	10	0.8532	4.5439	0.6597	6.5950	1.3384
	MSC	11	0.8675	4.4935	0.6355	6.3393	1.2520

**Table 3 foods-13-01863-t003:** Lettuce MLR modeled by characteristic wavelengths.

Indicators	Characteristic Wavelength Selection Methods	Number of Characteristic Wave Points	Lvs Number of Principal Factors	Calibration Set		Prediction Set		*RPD*
				$R_{C}$	*RMSEC*	$R_{P}$	*RMSEP*	
SSC	Raw	1360	15	0.8376	0.3144	0.8204	0.4152	1.7986
	CARS	22	18	0.8149	0.3382	0.8032	0.4318	1.7228
	UVE	468	17	0.8548	0.2976	0.8369	0.4007	1.8745
	SPA	17	15	0.7341	0.4123	0.7071	0.5297	1.3649
MC	Raw	1360	13	0.9247	0.8742	0.8169	1.3986	1.6892
	CARS	24	14	0.8697	1.1276	0.8410	1.3094	1.8447
	UVE	464	19	0.9174	0.9114	0.8223	1.3841	1.7134
	SPA	18	16	0.7468	1.5329	0.8041	1.4763	1.6053
VC	Raw	1360	16	0.7496	5.8320	0.7310	6.0328	1.5012
	CARS	21	18	0.7213	6.1108	0.7031	6.4914	1.4361
	UVE	434	17	0.8146	5.1817	0.7849	5.4597	1.6298
	SPA	23	19	0.7231	6.0986	0.5527	7.6327	1.2049

**Table 4 foods-13-01863-t004:** Lettuce PLSR modeled by characteristic wavelengths.

Indicators	Characteristic Wavelength Selection Methods	Number of Characteristic Wave Points	Lvs Number of Principal Factors	Calibration Set		Prediction Set		*RPD*
				$R_{C}$	*RMSEC*	$R_{P}$	*RMSEP*	
SSC	Raw	1360	21	0.8771	0.2718	0.8508	0.3839	1.9130
	CARS	25	23	0.8528	0.2955	0.8290	0.4075	1.8023
	UVE	437	20	0.8946	0.2587	0.8648	0.3792	1.9685
	SPA	17	9	0.7766	0.3968	0.7272	0.377	1.4383
MC	Raw	1360	14	0.9610	0.5767	0.8485	1.2654	1.8824
	CARS	33	8	0.8895	0.9532	0.8578	1.2174	1.9568
	UVE	445	15	0.9528	0.6330	0.8475	1.2626	1.8867
	SPA	16	14	0.7998	1.2544	0.8382	1.3451	1.7710
VC	Raw	1360	13	0.7923	5.3987	0.7888	5.4520	1.6285
	CARS	20	20	0.7478	5.8800	0.7682	5.6500	1.5714
	UVE	417	25	0.8345	4.9821	0.8047	5.2974	1.6689
	SPA	21	16	0.7549	5.8262	0.5843	7.1427	1.2365

**Table 5 foods-13-01863-t005:** Lettuce LSSVM modeled by characteristic wavelengths.

Indicators	Characteristic Wavelength Selection Methods	Number of Characteristic Wave Points	γ	$\sigma^{2}$	Calibration Set		Prediction Set		*RPD*
					$R_{C}$	*RMSEC*	$R_{P}$	*RMSEP*	
SSC	Raw	1360	12,546	4138	0.8704	0.2802	0.8106	0.4339	1.6789
	CARS	21	14,762	4684	0.7031	0.4068	0.7666	0.5042	1.4414
	UVE	487	9547	3621	0.8470	0.3023	0.7941	0.4505	1.6124
	SPA	16	11,436	3847	0.6973	0.4440	0.7861	0.4301	1.5499
MC	Raw	1360	9201	2548	0.8348	1.2345	0.8137	1.4682	1.6685
	CARS	29	9847	2998	0.8246	1.2897	0.8067	1.5078	1.6347
	UVE	383	8479	2136	0.8547	1.1049	0.8320	1.3746	1.7649
	SPA	18	10,345	3478	0.8147	1.3946	0.7698	2.1439	1.4581
VC	Raw	1360	9678	3147	0.7452	5.9123	0.7149	7.1948	1.4138
	CARS	26	10,698	3657	0.7346	6.0314	0.7039	7.3124	1.3887
	UVE	427	8524	2237	0.7982	5.4497	0.7586	6.7536	1.5248
	SPA	19	10,169	4079	0.7534	5.8017	0.7228	7.1146	1.4376

## Data Availability

The original contributions presented in the study are included in the article/supplementary material, further inquiries can be directed to the corresponding author/s.

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
