# Peer review of "Non-Destructive Inspection of Physicochemical Indicators of Lettuce at Rosette Stage Based on Visible/Near-Infrared Spectroscopy"

_foods, 2024, doi:10.3390/foods13121863_

Round 1

Reviewer 1 Report

Comments and Suggestions for Authors

The topic is interesting and up-to-date, but the biological side of the experiment is incompletely described in the manuscript. The introduction should be supplemented, the biochemistry part of the methodology should be rewritten, results need improvements. Please see attached manuscript.

Author Response

Thank you very much for the reviewer's valuable suggestions, according to your comments I have made a detailed revision of this paper, please refer to the attached file.

Reviewer 2 Report

Comments and Suggestions for Authors

The manuscript entitled: Non-destructive inspection of physicochemical indicators of lettuce at rosette stage based on visible/near-infrared spectroscopy, having authors: Wei Li, Qiaohua Wang and Yingli Wang, presents a study focused on hydroponically grown lettuce during the rosette stage and investigated the patterns of changes in indicators and spectral curves over time. By employing spectral preprocessing and selecting characteristic wavelengths, three models were developed to predict the indicators. In this study, lettuce was grown using hydroponic cultivation, and experiments were conducted to determine the SSC, moisture content, and vitamin C levels of lettuce at the rosette stage. The changes in internal indicators of lettuce during the rosette stage were analyzed based on the number of growth days. Additionally, non-destructive testing of physicochemical indicators of lettuce at the rosette stage was performed using visible/near-infrared spectroscopy. A portable device for visible/near-infrared spectroscopy was designed to achieve online non-destructive testing of physicochemical indicators of lettuce at the rosette stage.

The results showed that with the increase of head-forming days, the relative content of SSC initially increased, then decreased, and gradually increased again. The moisture content initially decreased and then slightly increased. The relative content of vitamin C decreased initially, sharply increased, then decreased again, and finally increased. Based on the visible/near-infrared spectroscopy curves of lettuce during the head-forming stage, it was observed that at 449 nm, 559 nm, 670 nm, 717 523 nm, 812 nm, and 901 nm, the spectral reflectance curves exhibited distinct peaks and valleys.

Author Response

We appreciate your professional comments on our articles and thank you for your positive comments and valuable suggestions to improve the quality of our manuscripts!

Reviewer 3 Report

Comments and Suggestions for Authors

This manuscript is devoted to studying the soluble solids content SSC, moisture content, and vitamin C levels of lettuce at the rosette stage.

Some corrections should be introduced:

Section 2.1.

lines 77-79, The phrase “hot water” is imprecise. The authors should provide the water temperature.

Please provide the conditions in the germination chamber (temperature, light intensity).

Please provide the conditions in the growth chamber (humidity, light intensity).

Authors should provide data on the devices used, including chambers (germination and growth): type, model, and manufacturer.

What nutrient solution was used? Please provide the composition.

The phrase "in the figure below" should not be used at work. Instead, it should say "in Figure 1".

Section 2.2.1.

The description of the method should not be an instruction manual.

Authors should specify what equipment was used (model, type, manufacturer).

Section 2.2.3.

What buffer reagent and MPA working solution were used? Please provide the composition.

Authors should specify what spectrophotometer was used (model, type, manufacturer).

Section 2.3.

Lines 159-166, The description of the method should not be an instruction manual.

Section 3.1.

Lines 213-220, The text should be in the "Materials and methods" section.

Section 3.6.

There are no results referring to Figure 10 in the text.

Author Response

Thank you for the valuable comments given by the reviewer, I have made the following changes in response to your comments:

Section 2.1.

lines 77-79, The phrase “hot water” is imprecise. The authors should provide the water temperature.

Increased water temperature conditions to 55°C

Please provide the conditions in the germination chamber (temperature, light intensity).

Please provide the conditions in the growth chamber (humidity, light intensity).

Authors should provide data on the devices used, including chambers (germination and growth): type, model, and manufacturer.

the conditions in the germination chamber:Temperature: 15°C,no light required.

the conditions in the growth chamber:Temperature: 20℃ Humidity: 60%RH Light: 30000Lx

The germination and growth chambers use instruments from the school academy.

What nutrient solution was used? Please provide the composition.

The variety of nutrient solution used was Hoagland's Nutrient Solution, and the nutrient solution was configured in the ratio of 1g of powder to 1L of water.

The phrase "in the figure below" should not be used at work. Instead, it should say "in Figure 1".

Changes have been made based on the comments you provided.

Section 2.2.1.

The description of the method should not be an instruction manual.

Authors should specify what equipment was used (model, type, manufacturer).

The instrument used was the Shen Guang WAY (2WAJ) Abelian refractor. The description of the method is the actual method of operation, which is a little bit the same as the operation manual.

Section 2.2.3.

What buffer reagent and MPA working solution were used? Please provide the composition.

Lettuce vitamin C content was measured using a vitamin C assay kit, which was produced by Shanghai Yuanye Biological Co. The buffer and MPA working solution were reagents in the kit.

Authors should specify what spectrophotometer was used (model, type, manufacturer).

The spectrophotometer use instruments from the school academy.

Section 2.3.

Lines 159-166, The description of the method should not be an instruction manual.

The description of the method is the actual method of operation, which is a little bit the same as the operation manual.

Section 3.1.

Lines 213-220, The text should be in the "Materials and methods" section.

The number of samples per experiment and the total number of samples are described here and can be placed here for greater clarity.

Section 3.6.

There are no results referring to Figure 10 in the text.

Figure 10 shows an index plot of the characteristic wavelength points of each metric selection on the original curve in the case of the optimal model, without citation.